# Underwater Target Signal Classification Using the Hybrid Routing Neural Network

**DOI:** 10.3390/s21237799

**Published:** 2021-11-24

**Authors:** Xiao Cheng, Hao Zhang

**Affiliations:** 1College of Information Science and Engineering, Ocean University of China, Qingdao 266100, China; chengxiao@stu.ouc.edu.cn; 2School of Physics and Electronic Engineering, Taishan University, No. 525 Dongyue Street, Tai’an 271021, China

**Keywords:** deep learning, convolutional neural network, signal classification, underwater acoustic environment

## Abstract

In signal analysis and processing, underwater target recognition (UTR) is one of the most important technologies. Simply and quickly identify target types using conventional methods in underwater acoustic conditions is quite a challenging task. The problem can be conveniently handled by a deep learning network (DLN), which yields better classification results than conventional methods. In this paper, a novel deep learning method with a hybrid routing network is considered, which can abstract the features of time-domain signals. The used network comprises multiple routing structures and several options for the auxiliary branch, which promotes impressive effects as a result of exchanging the learned features of different branches. The experiment shows that the used network possesses more advantages in the underwater signal classification task.

## 1. Introduction

In the complicated and volatile ocean, underwater acoustic target recognition is considered as the most challenging task, and the objective conditions of the marine environment seriously interfere with the recognition accuracy, which mainly includes transmission attenuation, multi-path effects, and ocean environmental noise. Underwater signals contain various forms, such as ships, marine organisms, oceanic turbulence and internal waves, etc. The instruments detect very faint sound waves, and there are various target acoustic features in the collected signals that are quite similar to the auditory system of humans used to recognize different spectrogram characteristics. In external circumstances, objective factors dramatically affect the ability to audibly interpret the signals, and it is difficult to precisely classify acoustic signals even by well-trained people.

In underwater sound signal identification, one of the most valuable subject areas is ship-type recognition. Signal classification technologies based on ship features have been gaining significantly increased interest over the years, resulting in a demand for an increase in complexity and precision of the corresponding state-of-the-art algorithms. There are various factors that present a significant obstacle to underwater acoustic signal recognition. The vast majority of methods include two steps: characteristic extraction and a recognition algorithm. There are many types of extraction method, such as the short Fourier transform (STFT) [1], low frequency analysis and recording (LOFAR) [2], Mel-frequency cepstral coefficient (MFCC) [3], and the detection of envelope modulation on noise (DEMON) [4].

The extracted characteristic programs that reflect underwater acoustic signals can be improved to some extent, and the losses of raw data are unavoidable in the processing, which leads to high computational complexity. The classifier techniques need to correspond with signal characteristic dimensions and spectrum features, which create difficulties for the designed algorithm. The characteristic selecting modes more or less result in the missing details of the raw signal; this determines how effective underwater acoustic identification algorithms will be for particular sound data. The recognition classifiers span traditional machine learning [5,6,7], the statistic approximation method [8], and matched field [9], which depend on critical prior knowledge and professional feature design, resulting in a dilemma in higher classification precision and greater operational efficiency.

Remarkable achievements have been obtained in the deep learning field, such as image vision, natural language processing, and voice identification [10,11,12]. DLNs have a distinct advantage in the classification and identification tasks, which benefits from the multilayer network architecture’s ability to extract nonlinear features. Signal recognition algorithms based on DLN have made considerable progress. There are more and more deep learning methods used in underwater acoustic signal recognition. The deep neural network (DNN) uses modified time–frequency characteristics as input, which more clearly expresses the outstanding center feature of ship samples. There are impressive classification results in the ship signal dataset, and it is due to the fact that the identification anchors based on the objective function are gained in the space distance [13]. The two dimensions of a convolutional neural network (CNN) excavates the ship signal characteristics in the spatial-temporal spectrum domain, which can weaken spectral fluctuation and prevent local minima [14]. An end-to-end learning network, called the auditory perception inspired deep CNN (ADCNN), is an efficient network architecture in signal feature extraction, and the method transform signals from the temporal domain to the frequency domain. The deep representation of raw signal can be separated; this method achieves satisfactory performances in underwater acoustic target classification [15]. The original signal data are input into the depthwise separable CNN (DSCNN) in the temporal domain. The desired identification effects suggest that the model inherits from the function of intra-class concentration, which isolates inter-class characteristics at the same time [16]. The deep-belief network utilizes the pre-processing approach of standard Boltzmann machine, the hidden middle layer of the clustering method, and the training optimization of parameter updating [17]. The underwater acoustic signal classifier adopts the convolutional recurrent neural network (CRNN), and the recurrent neural network (RNN), combined with a CNN, to acquire the different natures of sound characteristics, which further enhances the recognition effects by data augmentation [18].

Currently, CNN and RNN are only employed with a direct hierarchical overlay or a simple combined connection between them, which has not been optimized in terms of the network structure design. The underwater acoustic signals are seriously affected by the harsh underwater environment; therefore, it is important to mitigate these factors in the deep network design. The contributions of this paper are as follows:The hybrid routing network structure invests a simplistic format for the complex routing logic network. After several network units are overlaid structurally, the used network can generate multiple routing modes, which enhances the performance of extracted signal features.The network unit is arranged by the different branches. When the main branch remains unchanged, the auxiliary branch adopts three optional orientations, and the classification ability of the used network is furthered by exchanging advanced signal features in different branches. It enriches the extraction categories of signal features.The hybrid routing network was tested on real underwater ship signals. In the experiment, various multi-routing units were confirmed to illustrate the effectiveness of the used network, and there is also a comparative display of the different routing modes and the selection of auxiliary branches.

The remainder of this paper is organized as described below. The signal model is given in Section 2 along with the basic network structure. Section 3 explains the hybrid routing network structure form, and illustrates the multiple routing form and the optional auxiliary branches. Section 4 introduces the ship signal dataset, and the classification performance of the hybrid routing network provides experimental verification. Finally, Section 5 provides a summary of the paper.

## 2. Signals and Basic Network

### 2.1. Signal Description

In Figure 1, the original ship signals are one dimension of continuous data in the time domain, which is measured by signal length and sampling rate. In practice, however, the sound level felt by the human auditory system (HAS) is nonlinearly related with the frequency. Mel-frequency is inspired by HAS, which is more aligned with human ear requirements. Linear behaviors of mel-frequency are mainly below 1000 Hz, and there is a logarithm increase above 1000 Hz. It is tough to understand the difference between two very approximated frequencies expressed by the mel-frequency for HAS. Each frequency region is concerned only with the energy value, and the energies are added together in the region, which makes the regions more consistently recognizable by the human auditory system. In the same way, the filter energy builds up in the working scope, making the mel-spectrogram more separable than standard. During the ship signal processing, the mel-spectrum can be acquired by the energy spectrum in the filter output. The formula [19] is as follows
(1)$$Mel\left(i\right)==\sum_{\beta =h(i-1)}^{h(i+1)}G_{i}\left(\beta \right)\times {\left|T\left(\beta \right)\right|}^{2}$$

$G_{i}\left(\beta \right)$ is the filter output, i=1,2,⋯,I, the filter number is *I*. The fast Fourier transform (FFT) point number is β, β=1,2,⋯,E, and the total FFT frequency point is *E*. In the energy spectrum, the βth point energy is ${\left|T\left(\beta \right)\right|}^{2}$. The weight uses each filter output of the frequency range, and the energy spectrum of the corresponding frequencies is superimposed, which is only added up in the filter range. For a signal sequence, h(·) have *I* outputs. The several *I* output mel-filters are overlapped in time, which generate the mel-scale acoustic spectrum Mel(·) in Figure 2. Mel-spectrum data are fed into the used network to yield better classification results.

### 2.2. Basic Network Structure

For general machine learning algorithms, the quality of features has a vital impact on generalization performance. The features extracted by the manual design need to be backed up by professional knowledge for each kind of dataset; therefore, it is not easy to design good features manually. DLN isolates the proper features by itself, and there is no demand to design the special features for all sorts of datasets. Through multi-layer processing, DLN gradually converts the initial low-level features into high-level features, and completes the complex classification of the learning task by the trained network. These layers contain the information of the input data and the features flowing from one layer to another. At the upper layer, there are more features defined by the lower layer, which stands for the abstraction space output of each layer. The benefit of DLN comes from the fact that the deep structure is equal to a kind of factorization. Most randomly selected functions cannot actually be conducted, and many of them can be virtually expressed by the deep structure rather than the shallow structure. The existence of the network structure, which can be presented by depth, means that there is some corresponding structure of the potential representable functions to the source dataset. If there is no structure, it is impossible for the trained network model to be well generalized on a similar dataset. One-dimensional convolution is suited to natural language processing [20], while two-dimensional convolution and three-dimensional convolution have more extensive applications in computer vision [21]. Two-dimensional convolution is shown in Figure 3a. The image data are composed of two-dimensional data of 16×16, and 3 represents the depth of image data. The middle yellow cube represents the range of convolution operation on the raw image, where the convolution kernel is 3×3×3. After traversing the complete image data, the results are obtained by the image data of 14×14×1 on the right. The deeply separable convolution form divides the traditional convolution operation into two steps, shown in Figure 3b. The first is the deep convolution process. The input is convoluted by the 3×3×1 convolution kernel without changing the depth of image data, which corresponds to the image data of 14×14×1. These image data are stacked together to create the image data of 14×14×3. The second step is the point-by-point convolution, and the 1×1 convolution kernel traverses every point of the 14×14×3 image data to retrieve the 14×14×1 convolution consequence.

The deep learning network typically molds a large number of stacked network layers to improve the learning ability and achieve higher classification results. The multi-layer network makes it easy to implement the deep network model. When the trained network model is generalized to a similar dataset, some network hyperparameters need to be adjusted, leading to the scalability problem of sparse network structure, which hinders the application in the UTR task. To ameliorate the problem, the balanced structure is considered by the form of the multi-route sparse network structure, which is combined with the short-connection network method [22]—it is equivalent to shortening the depth of the network, as shown in Figure 4a, which does not affect the extracting ability of the network. The formalism removes the limitation that a layer output can only be constituted as the next layer input, and allows the output of a layer to be the farther separation and extraction as the input of the next multiple layers. The short-connection network components organize a new structure as shown in Figure 4b.

## 3. Hybrid Routing Network Architecture

For gaining abundant classification features, the most general style of multi-route sparse network architecture is the intricate structure of the sparse fragmented network, which can extract the diversiform signal data; it is a valid model to improve the classification performance, whereas the sparse fragmented architecture has a poor adaptability to the signal dataset, especially in the various number of signals. Specifically, to benefit from the architecture, the new network is created, and named as the hybrid routing network—it further strengthens the ability to acquire the deep relating information of signals, and is a suitable method to handle the various input signal data.

The basic composition of the hybrid routing network is shown in Figure 5. The hybrid routing network can achieve the effect of the sparse fragmented network to extract a variety of signal features, and it is dissimilar to the sparse fragmented network to reduce the training parallelism degree, such as ResNeXT [23] and Nasnet automatically generated by AutoML [24]. The sparse fragmented network tends to adopt the complex structure, where there are multifarious small convolutions and pooling operations in the layer structure, which leads to the complexity of the network structure, reduces the efficiency of the model, and slows down the training speed. The hybrid routing network can solve these problems by keeping the unchanged structure in the branches. In order to realize the capacity and efficiency of the ideal model, the key is how to maintain a large number of branches with the same width. In this manner, there are neither dense convolutions nor too many Add operations.

The input layer is fed by the original signal data. The gate recurrent unit (GRU), two-dimensional convolution (Conv_1), and MaxPooling pre-process the signal data. There are 512 cells in GRU, and the convolution kernels are 3×3 in other two pre-processing layers. The overall network structure is constructed by superimposing the multi-routing units (red solid wireframe). At the beginning of each unit, the network input is divided into different branches. The main branch $l_{0}$ consists of seven basic layers, including group convolution (GConv) with the convolution kernel of 1×1, three dropouts, ReLU activation function (ReLU), depthwise convolution (DepthConv_1) with the convolution kernel of 2×2, two-dimensional convolution (Conv_2) with the convolution kernel of 1×1. The three branches that can be selected on the auxiliary path correspond to $l_{1}$,$l_{2}$,$l_{3}$, respectively (red dotted wire frame). The auxiliary branch $l_{1}$ consists of five basic layers, including DepthConv_2 that has the 3×3 convolution kernel, two dropouts, Conv_2, and ReLU. The auxiliary branch $l_{2}$ includes the average pooling (AveragePooling), and the auxiliary branch $l_{3}$ is a directly connecting link. The track restructure (TrackRestr) is the feature exchange operation between the different branches, and the concatenate layer (Concat) splices the extracting data from the different branches for continuing to learn in the next network unit. The structure is realized through the superposition of the multi-routing units. The corresponding formula is as follows:(2)$$L=\sum_{j=1}^{J}\left[l_{0}+\sum_{u=1}^{U}\zeta \left(l_{u}\right)\right]$$where *j* represents the superimposed units, j=1,2,⋯,J, and *J* represents the maximum number of superimposed units. $l_{0}$ represents the main branch, ζ(·) represents a selection function to the auxiliary branch, *u* represents the alternative mode of different auxiliary branches, u=1,⋯,U, and *U* represents the total number of auxiliary branches. The *j* layer can choose any optional auxiliary branch required from 1 to *U*. L represents the final network structure.

The different branches exchange the signal features are shown in Figure 6. Branches (black solid wire frame) represent the multi-routing path from 1 to *p*th. Packets are the signal features, and tracks (purple solid wire frame) are the selected range for the features exchange of each branch at a certain percentage. It is provided to the next unit for further learning by rotating and changing a certain proportion of features between different branches, which can avoid the limitations of different branches to pledge the rich signal features. The general convolution operates on all input feature maps, which is the full-track convolution. It is the track dense connection, which means that the convolution is performed on all tracks. In fact, the 1×1 convolution in ResNeXt basically takes up more than 90% of the multiply-add operations. Xception [25], MobileNet [26] also use a similar convolution; these algorithms also adopt DepthConv, which is actually a special convolution. Some packets make up a track, and each packet has only one feature map. The DepthConv amounts to adding a barrier to remove unnecessary data, thus reducing the computation amount. The disadvantages brought by the sparse fragmented network are solved by restructuring the convolution tracks, and it means that the hybrid routing network is a track sparse connection. When the GConv layer is stacked, the next problem is that the feature maps between different branches do not communicate, which is similar to dividing mutually irrelevant branches. The extracted features go their own way, and will reduce the feature extraction capability of the network.

The information contained different branches may be similar in the same packet. If there is no track exchange, the learned features will be very limited. If some tracks are exchanged after different branches, and the learned information can also be exchanged. Each packet has more information and more features can be extracted. To achieve the features of all other packets in each branch, the form is conducive to better results. For this reason, Xception and MobileNet are equipped with the serried 1×1 convolution, which ensures the information exchange between diverse packets of feature maps after the convolution operation, referred to as the restructuring conversion. The solution guarantees that the next convolution input comes from diverse packets so that the information can be transferred between diverse packets. Normally, the restructuring conversion is not random, but evenly disrupted, which is more beneficial for the sharing of the learned feature between the distinct manifestation of tracks; this only requires the simple dimension transformation and transposition to achieve an even restructure, which is simple and easy to operate. The two-dimensional feature matrix corresponding to each branch is $W_{1}$, $W_{2}$, ⋯, $W_{p}$, and the selected feature range percentage is δ. The features involved in the exchange are(3)$$\begin{array}{c} V_{1}=\delta W_{1}\\ V_{2}=\delta W_{2}\\ \cdots \\ V_{p}=\delta W_{n} \end{array}$$after the first network unit learns, the proportionally selected initial matrix is(4)$$\left[V_{1},V_{2},V_{3},\cdots ,V_{p-1},V_{p}\right]$$the rotating and changing operation from 1 to (p−1)th is(5)$$\left[V_{p},V_{1},V_{2},\cdots ,V_{p-2},V_{p-1}\right]$$(6)$$\left[V_{p-1},V_{p},V_{1},\cdots ,V_{p-3},V_{p-2}\right]$$(7)$$\begin{array}{c} \cdots \\ \left[V_{2},V_{3},V_{4},\cdots ,V_{p},V_{1}\right] \end{array}$$to complete the exchanging operation, these features are connected into the feature sequence of the first multi-routing unit by Concat, which can be transmitted to the next multi-routing unit for further learning. When the above stage is completed, the global map pooling (GlobalMapPooling) is intended to reduce the feature map size to 1×1, and finally the fully connected layer (Dense) outputs the ship type predictions.

## 4. Experiment

### 4.1. Training Setting and Ship Signal Dataset

The training setting of the used network is a batch size of 128; the optimizer selects the stochastic gradient descent (SGD) with momentum = 0.9, decay = $5\times 10^{-4}$, and learning rate = 0.05. To elevate the extension ability of the trained network, the early stopping technique uses a patience of 5.

In real shallow sea water, various forms of ships generate the signal dataset. Human disturbance and underwater environmental impacts are contained in the target signal dataset. The ship acoustic data are collected by an hydrophone, which was laid under water at 144 m. The approximate distances range from 50 m to 150 m, and the signal sampling frequency is 32 kHz. The classes of signal recordings are well annotated, which are conveniently applied to automatic identification algorithms. There are approximately 90 samples in each ship target signal type, which range from 9∼682 s in sample duration.

### 4.2. Classification Performance

Figure 7 shows the classification performance with different routing forms. There are the superposition of three units in the network structure. (X,X) represents the method of the auxiliary selection corresponding to the intermediate overlay units. Other forms of (X,X,X) have similar meanings, and *X* is the different choice of $l_{1}$, $l_{2}$, or $l_{3}$. δ is the percentage of the selected packet range as a track for the exchange between auxiliary branches. In the 12 different hybrid routing forms, 12 ship types can be effectively identified. When δ is 20%, there are similar classification results in the different routing forms. With the increase in δ, the classification ability differs from the the selection mode of auxiliary branches. The form of $(l_{1},l_{2},l_{3}),\delta =100\%$ increases by approximately 3,93%, 3.40%, and 4.09% compared to the three forms of $\left(l_{1},l_{2}\right)$, $\left(l_{1},l_{3}\right)$, and $\left(l_{2},l_{3}\right)$ at δ=20%, which has a distinct advantage over other routing forms. The presence of different routing forms turns out to be the sheep herd performance. $(l_{1},l_{2},l_{3}),\delta =100\%$ slightly improves 3.27%, 2.58%, and 2.44% than the other three routing forms of $\left(l_{1},l_{2}\right)$, $\left(l_{1},l_{3}\right)$, and $\left(l_{2},l_{3}\right)$ at δ=60%. $(l_{1},l_{2},l_{3}),\delta =100\%$ is better around 1.85%, 2.40% than compared to $(l_{1},l_{2},l_{3}),\delta =60\%$, $(l_{1},l_{2},l_{3}),\delta =20\%$, which performed better at 2.13%, δ=100% of the other three routing forms on average. When the sufficient exchange of the signal data are provided to the used network, more high dimensional signal features are extracted, which helps to identify ship types. The further addition of routing branches did not improve classification accuracy due to the fact that the hybrid routing network can more effectively extract the signal features by the full exchange of tracks, and the advantageous classification effect can be achieved under the $(l_{1},l_{2},l_{3}),\delta =100\%$ form; this explains the fact that the hybrid routing network is an efficient method for ship-type classification.

Figure 8 shows the convergence performance for the used network during the training and validating process, which involves different multi-routing units. In Figure 8a, the training signal dataset send data across the used network to obtain the training losses by the categorical cross-entropy. In Figure 8b, the validating signal dataset verify the trained network to obtain the validating losses, and the validating results are obtained by the same loss function, such as the training losses. In horizontal axis, the number of epoch is the learning number of the training process, which represents a whole training or validating dataset that moves through the network and returns once. At the beginning, the six kinds of the used network structures show a rapid convergence in the iterative procedure of the top five. When the multi-routing unit is six, the training loss obtains the best results, which has a longer epoch number. There is a similar convergence performance for the other five, which is higher than the multi-routing unit of six by an average of more than 0.004. As the multi-routing unit number increases, the epoch number displays a declining trend.

The validing loss of four multi-routing units is larger than that of two multi-routing units and three multi-routing units, and it is due to the fact that the validing dataset is under-fitted. Although the epochs of four multi-routing units are larger than the previously mentioned units, they cannot learn more hidden signal distinguishing features, resulting in larger validating losses. With the increase in multi-routing units, the ability to obtain signal hidden features is further improved, which can effectively reduce validing losses and achieve better classification results. Six multi-routing units and one multi-routing unit are partially coincident in the initial validing process. One multi-routing unit has a small number of layers, and it is impossible to obtain deeper signal classification information, which stops learning after 13 epoches. Six multi-routing units can dig out the deep distinguishing features of signals, and more epoches can effectively improve the classification effect. The training process with six multi-routing units and one multi-routing unit are partially coincident in the initial validating process. One multi-routing unit has a small number of layers, and it is impossible to obtain deeper signal classification information, which stops learning after 13 epochs. Six multi-routing units can dig out the deep distinguishing features of signals, and more epochs can effectively improve the classification effect, which shows that the used network can work effectively to learn the signal data features. The validating process is compared with the training process, and there is an approximate convergence tendency. The validating process is not the smooth course similar to the training process, and it is due to the fact that the probability distribution of the validating dataset is not entirely consistent with those of the training dataset, which can productively fulfill the validation of the trained network. In the validating process, the epoch number is also decreasing as the amount of multi-routing units increases, which is similar to the training process. It shows that the used network can effectively handle the ship acoustic signals by various multi-routing units.

In Figure 9, the used network is compared with DNN [13], ADCNN [15], DepthCNN [16], CDBN [17], and CRNN [18]. HRNet is the hybrid routing network method. In Figure 9a, HRNet has the better classification result than other networks in the epoch range of 0 and 6, and the recognition performance of CRNN is close to the used network, which significantly outperforms DepthCNN and CDBN. ADCNN and DNN have almost the same classification result when compared to HRNet, and the former are 4.96% than the latter. From the sixth epoch to the thirty-second epoch, the results generated by CRNN, ADCNN, and DNN share a great deal of similarity, which are 3.18% smaller than HRNet on average. At the scope from 33 to 43, HRNet is substantially higher than other networks in the classification rate, which is around 0.98%, 2.43%, and 3.17% than CRNN, ADCNN, and DNN on average, respectively. This shows that HRNet is structurally superior and obtained a more advanced classification of signals. In the validating accuracy process, there is an impressive melioration in the classification effect of all networks in Figure 9b. The reason for vibration is that the training dataset is not completely corresponding to the validating dataset. HRNet shows impressive performance compared to the five other networks. ADCNN and DNN show an approximate tendency toward the classification effect; they are less effective than HRNet, which is 4.46% and 3.67% better than ADCNN and DNN, respectively—this is due to the fact that the hybrid routing network structure enriches the feature extraction of signals, which gain a better performance than CRNN, ADCNN, DNN, DepthCNN, and CDBN. In the classification of 11 ship types, HRNet recognizes more than 95% in Figure 10, and the accuracy of the Tugboat can also achieve 92%, which proves the target recognition ability of HRNet.

## 5. Conclusions

In this paper, we analyzed the ship classification of the hybrid routing network in an ocean sound environment. The particular conditions found underwater make it difficult to achieve a high classification accuracy.

The used network with the multiple routing forms and the optional auxiliary exchanging branches contributes to extracting a plenty of signal features and further boost the classification performance. The used network with the multiple routing forms contributes to extracting plenty of deep signal features, which effectively boost classification performance. The optional auxiliary exchanging branches are an effective mechanism in signal feature mining, which enhances the learning ability of the used network.

Through the experimental analysis of the used network, we show that the network structure design is one of the best means to improve the recognition effect. The features learned in the used network can also further enhance the classification effect. It is of more practical significance to raise the efficiency of the underwater target recognition system on harsh terms. The used network method can also be extended to the other signal classification scene. In the future, we will study the hybrid routing method for the communication signal classification in an underwater acoustic environment.

## Figures and Tables

**Figure 1 sensors-21-07799-f001:**
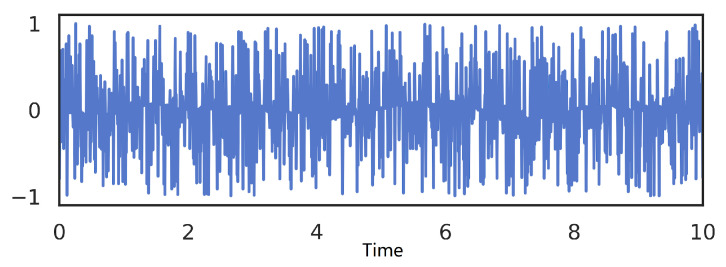
The underwater acoustic ship signal.

**Figure 2 sensors-21-07799-f002:**
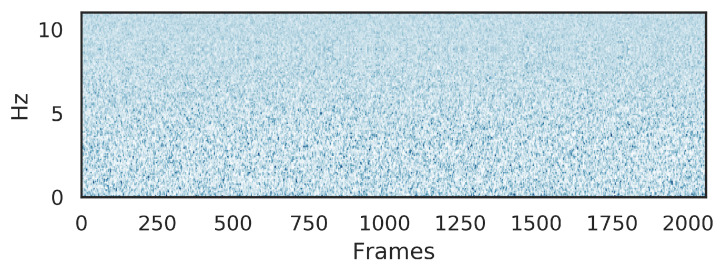
The mel-spectrum of the underwater acoustic ship signal.

**Figure 3 sensors-21-07799-f003:**
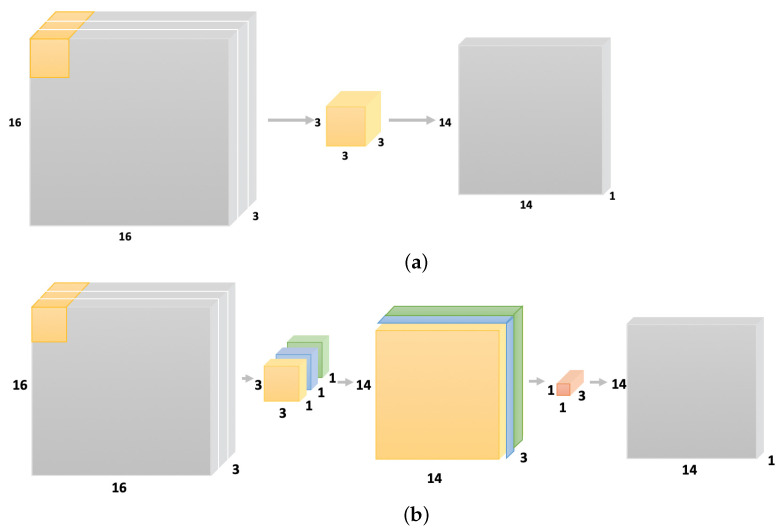
Traditional conventional and separable convolutional forms. (**a**) Traditional convolution. (**b**) Separable convolution.

**Figure 4 sensors-21-07799-f004:**
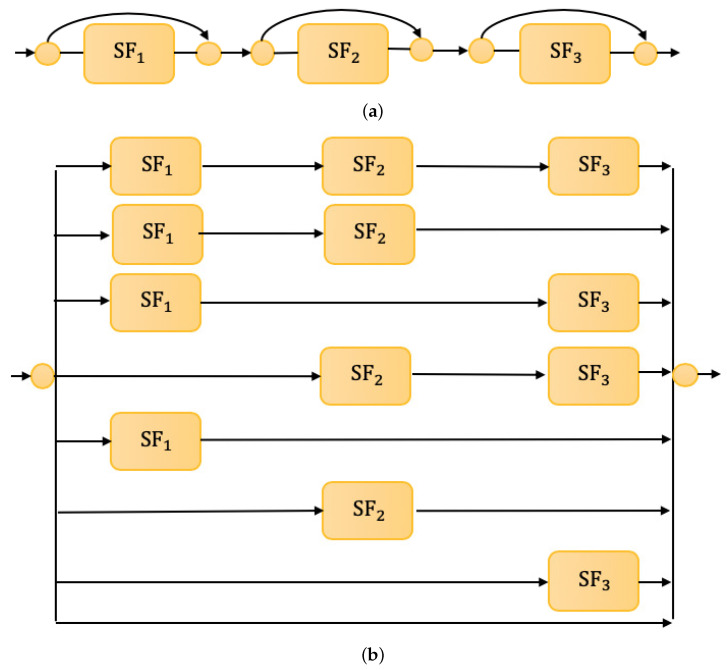
Multi-route sparse network architecture. (**a**) Short-connection network structure. (**b**) Equivalent multi-route network structure.

**Figure 5 sensors-21-07799-f005:**
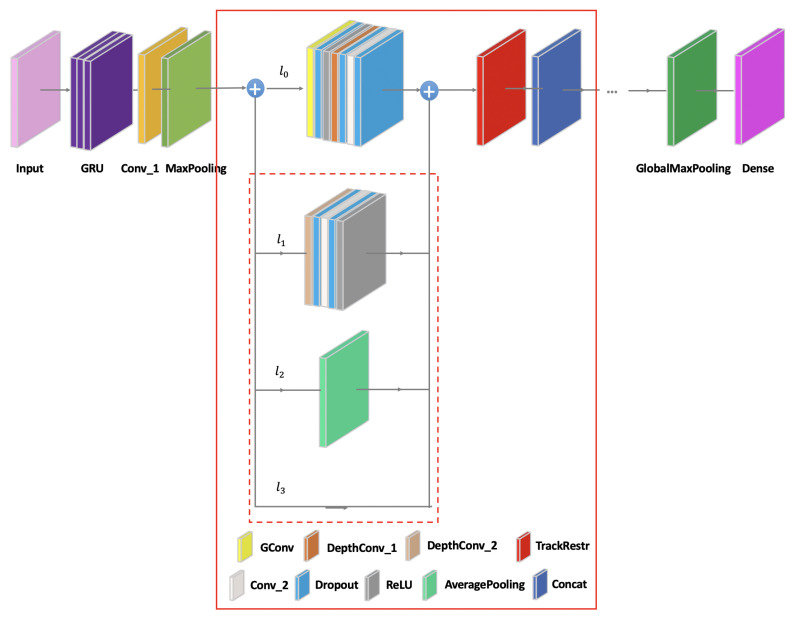
The hybrid routing network structure.

**Figure 6 sensors-21-07799-f006:**
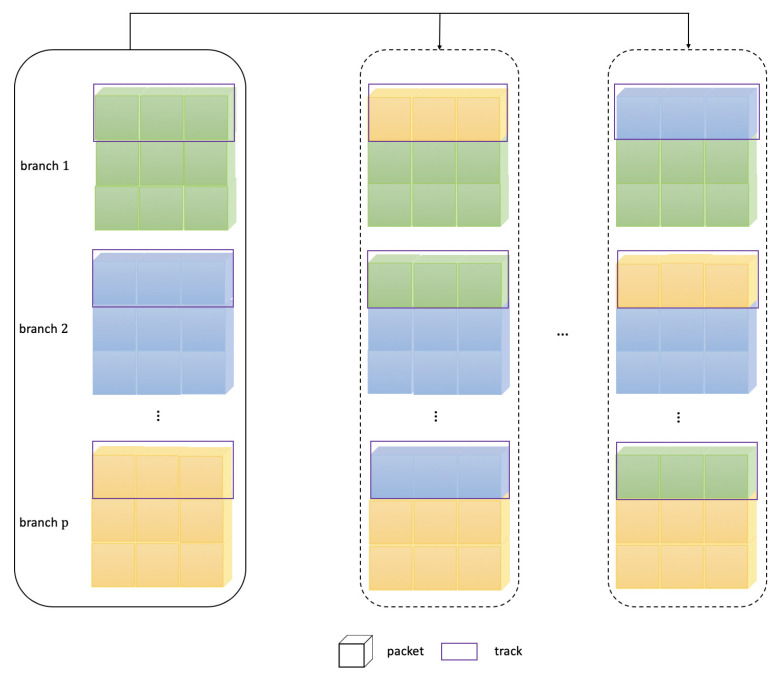
The feature exchanges in the different branches.

**Figure 7 sensors-21-07799-f007:**
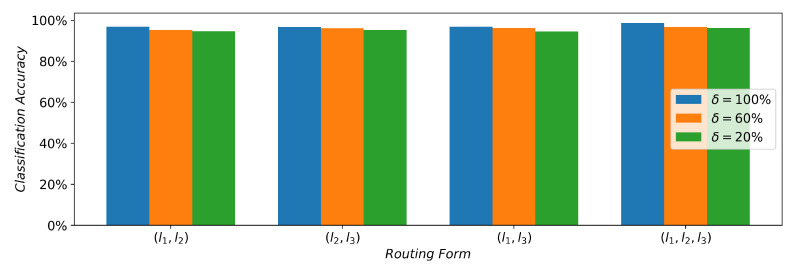
Classification accuracy with different routing forms and tracking exchanges.

**Figure 8 sensors-21-07799-f008:**
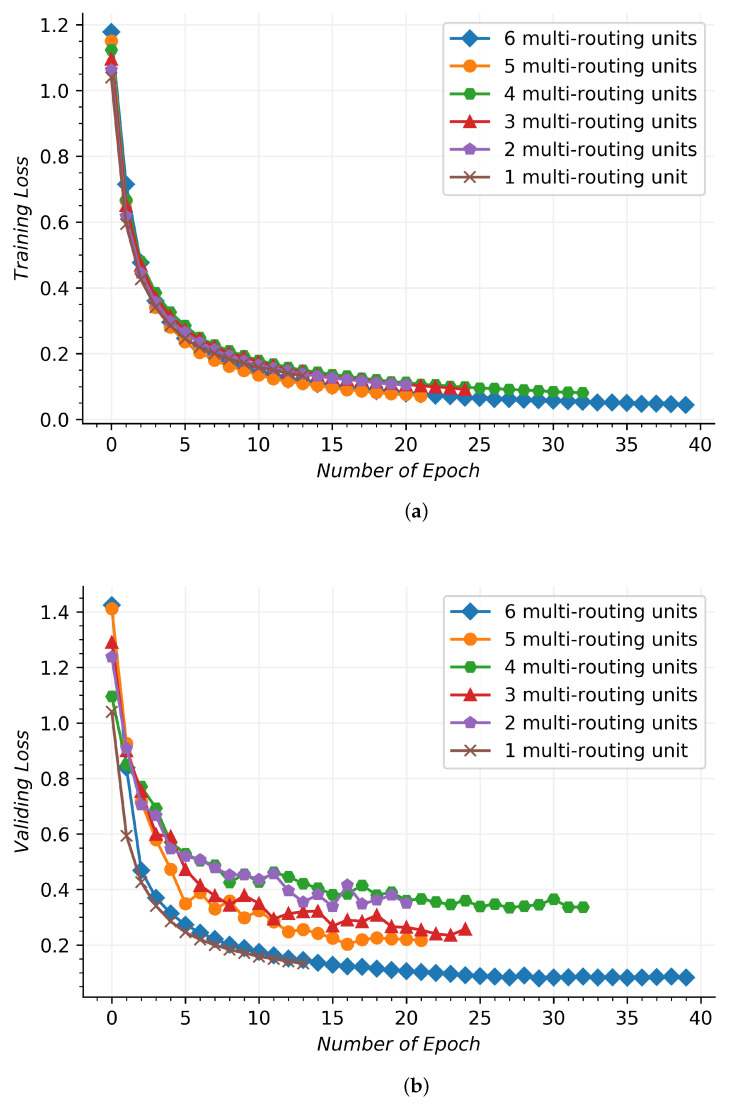
Training and validating process in the hybrid route network. (**a**) Training process. (**b**) Validating process.

**Figure 9 sensors-21-07799-f009:**
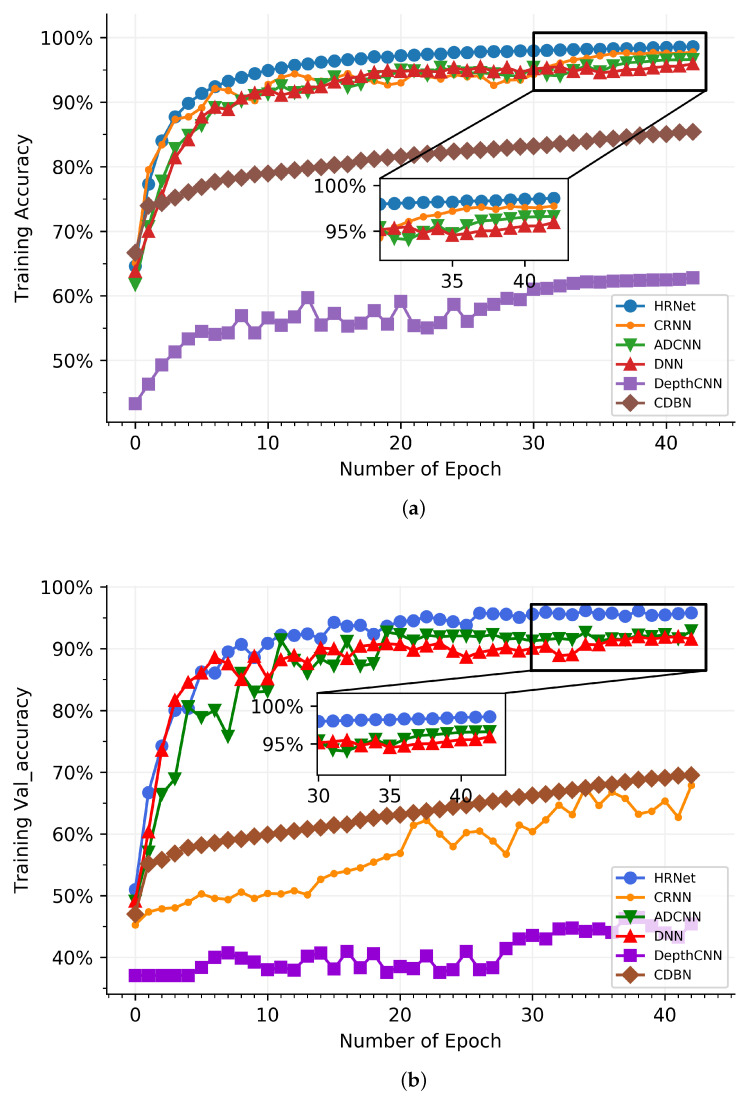
Classification results between different networks. (**a**) Training accuracy. (**b**) Validating accuracy.

**Figure 10 sensors-21-07799-f010:**
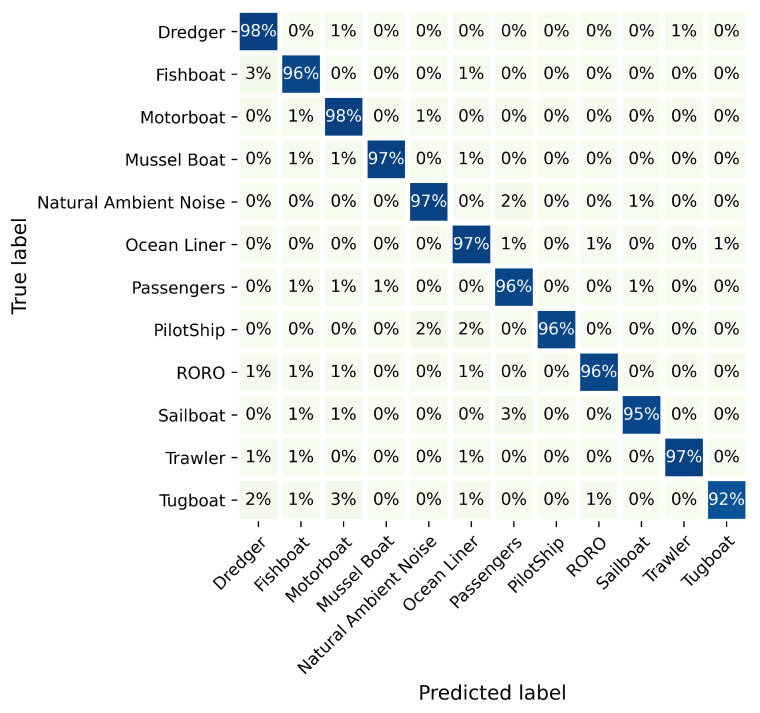
Ship type classification results.

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
