# Peer review of "Underwater Target Signal Classification Using the Hybrid Routing Neural Network"

_sensors, 2021, doi:10.3390/s21237799_

Round 1

Reviewer 1 Report

This paper presents a novel deep learning method with a hybrid routing network, which can abstract the features of time-domain signals. The used network includes multiple routing structure and several options for the auxiliary branch, and there are effects by exchanging the learned features of different branches. Consistently, the authors describe the signal model along with the basic network structure, explain the hybrid routing network structure form, illustrating the multiple routing form and the optional auxiliary branches. Finally, it is considered the ship signal dataset, and the classification performance of the hybrid routing network gives the experimental verification.

Some shortcoming and missing of the paper are the following:

  1. It should be pointed the reference to formula (1).
  2. How could be explained that in Fig. 8b, the curve corresponding to the case of 4 multi-routing units is significantly higher than the curve corresponding to the case of 3 multi-routing units and practically coincides with the curve corresponding to the case of 2 multi-routing units? At the same time, the curve, corresponding to the case of 1 multi-routing units, coincides with the curve corresponding to the case of 6 multi-routing units.
  3. Conclusions is very poor and should be significantly expanded for account of the results, obtained in the paper.

Reviewer 2 Report

The work presented in this paper is an interesting analysis of a novel deep learning method using a hybrid routing network applied to the ship classification in the ocean sound environment. The experimental work has evaluated the classification results between different networks in comparison to the new approach and the results show that the hybrid routing network obtains the most advanced classification information of signals.

The paper is well presented, but a minor spell check and typo are required.  

4. experiment should be 4. Experiment

4.1. Training Setting and ship Signal Dataset should be 4.1. Training Setting and Ship Signal Dataset

Round 2

Reviewer 1 Report

The paper can be published in the revised form.